# Another Look at Obesity Paradox in Acute Ischemic Stroke: Association Rule Mining

**DOI:** 10.3390/jpm12010016

**Published:** 2021-12-29

**Authors:** Pum-Jun Kim, Chulho Kim, Sang-Hwa Lee, Jong-Hee Shon, Youngsuk Kwon, Jong-Ho Kim, Dong-Kyu Kim, Hyunjae Yu, Hyo-Jeong Ahn, Jin-Pyeong Jeon, Youngmi Kim, Jae-Jun Lee

**Affiliations:** 1Institute of New Frontier Research Team, Hallym University College of Medicine, Chuncheon 24252, Korea; pumjun4093@gmail.com (P.-J.K.); bleulsh@naver.com (S.-H.L.); deepfoci@hallym.or.kr (J.-H.S.); gettys@hallym.or.kr (Y.K.); poik99@hallym.or.kr (J.-H.K.); doctordk@naver.com (D.-K.K.); yunow@hallym.or.kr (H.Y.); 21hobs@gmail.com (H.-J.A.); jjs6553@daum.net (J.-P.J.); kym8389@hanmail.net (Y.K.); iloveu59@hallym.or.kr (J.-J.L.); 2Department of Neurology, Chuncheon Sacred Heart Hospital, Chuncheon 24253, Korea; 3Department of Anesthesiology and Pain Medicine, Chuncheon Sacred Heart Hospital, Chuncheon 24253, Korea; 4Department of Otorhinolaryngology-Head and Neck Surgery, Chuncheon Sacred Heart Hospital, Chuncheon 24253, Korea; 5Chuncheon Artificial Intelligence Center, Chuncheon Sacred Heart Hospital, Chuncheon 24253, Korea; 6Department of Neurosurgery, Chuncheon Sacred Heart Hospital, Chuncheon 24253, Korea

**Keywords:** infarction, risk factors in epidemiology, outcome research, association rule mining, body mass index

## Abstract

Though obesity is generally associated with the development of cardiovascular disease (CVD) risk factors, previous reports have also reported that obesity has a beneficial effect on CVD outcomes. We aimed to verify the existing obesity paradox through binary logistic regression (BLR) and clarify the paradox via association rule mining (ARM). Patients with acute ischemic stroke (AIS) were assessed for their 3-month functional outcome using the modified Rankin Scale (mRS) score. Predictors for poor outcome (mRS 3–6) were analyzed through BLR, and ARM was performed to find out which combination of risk factors was concurrently associated with good outcomes using maximal support, confidence, and lift values. Among 2580 patients with AIS, being obese (OR [odds ratio], 0.78; 95% CI, 0.62–0.99) had beneficial effects on the outcome at 3 months in BLR analysis. In addition, the ARM algorithm showed obese patients with good outcomes were also associated with an age less than 55 years and mild stroke severity. While BLR analysis showed a beneficial effect of obesity on stroke outcome, in ARM analysis, obese patients had a relatively good combination of risk factor profiles compared to normal BMI patients. These results may partially explain the obesity paradox phenomenon in AIS patients.

## 1. Introduction

Stroke is a leading cause of death and disability globally [1]. Obesity is a known risk factor for cardiovascular disease (CVD) [2]. Reports also suggest that obesity may be protective for CVD severity or outcome [3,4,5]. Though most population-based cohort studies have shown that obesity increases the risk of stroke [6,7,8,9], obesity has been associated with better outcomes among patients with acute ischemic stroke (AIS) [10,11,12,13,14,15,16]. Khan et al. evaluated associations between BMI and lifetime CVD risk and mortality among a population-based cohort without established CVD at baseline [17]. Observing more than 3 million person-years, they found that being overweight was associated with earlier development of CVD, and obesity was associated with reduced longevity and cardiovascular survival. In addition, obesity increased the risk of hypertension, diabetes, and dyslipidemia. There is still insufficient explanation to determine whether the relationship between obesity and CVD outcome analyzed using logistic regression is beneficial or not.

The association rule mining (ARM) algorithm was introduced for market basket analysis by Agrawal et al. and has identified significant association patterns within a variety of settings [18,19,20,21,22,23]. ARM is used for explanatory data visualization to summarize certain concurrent combinations of dependent factors (important association rules) with a specific condition in a non-hierarchical fashion [19,24]. Moreover, ARM helps to discover important relationships among complex phenomena [18,19]. Therefore, we aimed to verify the obesity paradox with logistic regression and provide a new perspective through the ARM method on the relationship between obesity and ischemic stroke outcome.

## 2. Materials and Methods

### 2.1. Participants

We obtained data from the prospective stroke registry of our hospital, the detailed information of which has been described previously [25]. Briefly, patients were admitted to the hospital within 7 days after the onset of focal neurologic deficits (2012–2019). Patients included in the study were identified as having relevant acute lesions on diffusion MRI. Patients with cerebral hemorrhages were excluded from the study. During the data capture period, procedures affecting the prognosis of patients, such as extending the time window of intravenous thrombolysis and endovascular treatment, were reflected in all included patients (Appendix A). We collected demographic data and accessed data for clinical and laboratory findings for patients admitted with AIS using a standardized web server, which included central coordinator requests for regular audits and amendments of the data.

#### 2.1.1. Measurement of Body Mass Index

Height and weight were recorded by stroke nurses immediately after hospitalization for stroke. BMI was calculated as weight in kilograms divided by the square of height in meters (kg/m^2^). Patients were grouped as underweight (<18.5 kg/m^2^), normal weight (18.5–22.9 kg/m^2^), overweight (23–24.9 kg/m^2^), and obese (≥25 kg/m^2^) according to the Asian Pacific World Health Organization criteria [26].

#### 2.1.2. Covariates

Vascular risk factors were defined based on our previous report [25]. ARM generally assesses possible associations between discrete variables, and the algorithm can be easily applied to categorical variables. Therefore, we used the following categorical variables for the elucidation of risk profiles and the underlying mechanisms of stroke:Hypertension was defined as blood pressure ≥140/90 mmHg in more than two consecutive readings or taking antihypertensive agents;Diabetes was defined as fasting blood glucose ≥126 mg/dL, random blood glucose readings ≥200 mg/dL with relevant diabetic symptoms or glycated hemoglobin ≥6.5% [27];Hyperlipidemia was defined as total cholesterol ≥240 mg/dL or taking lipid-lowering agents [27];Current smoking was defined as smoking within 6 months prior to the study;Stroke subtypes were defined as cardioembolism and non-cardioembolism;Stroke severity was categorized as a score of the National Institute of Health Stroke Scale score (mild: 0–5, moderate: 6–14, and severe: >14) [28];Thrombolysis was defined as patients receiving intravenous or intra-arterial thrombolytic agents, or mechanical thrombectomy;Patient’s Age (years) was categorized as years < 54, 55 ≤ years, < 65, and years ≥ 65 [28].

#### 2.1.3. Study Outcomes

Stroke physicians or certified nurses prospectively assessed patients’ modified Rankin Scale (mRS) score as a 3-month functional outcome measure when the patients visited the outpatient clinic or via a telephonic interview. The poor functional outcome was defined as an mRS score of 3–6.

### 2.2. Statistical Analyses

We compared the baseline differences for independent variables between good and poor outcome groups using the χ^2^ test or Student’s *t*-test, as appropriate. Univariate and multivariate binary logistic analyses were performed to assess predictors for poor outcomes at 3 months. Independent variables with a two-sided *p*-value < 0.05 in univariate analyses were assessed by multivariate analyses.

### 2.3. Association Rule Mining

The frequent pattern growth (fp-growth) is one of the ARM algorithms used to evaluate associations for each variable, and to find important linkage patterns using the quantitative parameters of support, confidence, and lift [29]. The definition and formula of these quantitative parameters are stated in Table 1. To understand these parameters, let X and Y as events in the real world, then p(X) and p(Y) are the probabilities that these events will occur. Here, the sup (X→Y) means the probability that X and Y occur at the same time. The conf (X→Y) is a conditional probability such that the likelihood of Y occurs given event X. The last parameter lift (X→Y) is defined as a weighted probability of confidence, p(Y|X)/p(Y), to verify whether event X and Y are mutually dependent (when the lift is equal to one, then X and Y are independent).

The fp-growth algorithm is emphasized by its efficiency of a reduced number of calculations compared to the apriori algorithm, due to its tree-like structure [29]. For illustration of this algorithm, let T denote set of transactions {$t_{1}$, $t_{2}$, …, $t_{n}$} and list of all items, I = {$i_{1}$, $i_{2}$, …, $i_{m}$} (Table 2). First, reorder items in each $t_{j}$ according to the support ($i_{k}$) in descending order for all *k* in [1, m]. In these ordered transactions, the less frequent items which are smaller than user-defined minimum support are excluded for fewer calculations later. As a second step to grow an fp-tree, position and connect the items in the initial transaction, $t_{1}$, starting with a root node (null node), as in Figure 1. Each time, when items are added to the tree, increment the count of these item nodes by one. Third, for the following transaction, if the part of the sequence from the first item overlaps with one of the antecedent transactions, then the overlapping path of these items has to be merged, and the rest of the items are connected after the merged node. However, if there is no identical order of prefix, the new transaction follows the second step by connecting a sequence of items to a null node. Last, since the unit item is also a candidate for being a frequent pattern, the dashed lines in the fp-tree are drawn to denote the same unit items in the different branches of the tree. With a fully-grown tree by those steps, the candidates of frequent patterns are considered through minimum support and confidence. For example, in Figure 1, when the minimum support τ equals two, the sequence {null, $i_{1}$, $i_{3}$} (equality with {$i_{1}$, $i_{3}$}) is the only candidate satisfying the lower bound, τ, which means this sequence is a frequent pattern (rule) of the transaction data. Other candidate patterns through the fp-growth algorithm can also be considered through less conservative minimum supports.

In the resulting rules in this paper, pruning step, a rule XY is redundant if ∃X^*^⊂X and confidence(X^*^→Y) ≥ confidence(X→Y), is applied to drive a more generalized frequent pattern without redundant rules [30]. Since the goal of ARM is to find the rule of associations between categorical variables, the age variable was grouped as age < 55, 55 ≤, age < 65, and age ≥ 65 in our analysis. The moonBook (version 0.2.3), arules (version 1.6.4), and rCBA (version 0.4.3) packages for R software [R Foundation for Statistical Computing (version 3.6.3)] were used to perform the binary logistic regression and ARM algorithm. The exact R code for ARM using the fp-growth algorithm was shown in Appendix A.

## 3. Results

In total, 2580 AIS patients (mean age 68.1 ± 13.0 years, 59.1% males) were included in this study (Table 3). The proportions of underweight, normal weight, overweight, and obese patients were 4.6%, 34.8%, 26.4%, and 34.1%, respectively. The proportions of overweight and obese patients were higher in the good outcome group compared to the poor outcome group (*p* for χ^2^ trend <0.001). Participants with a poor outcome were more likely to be older, have severe stroke severity, to present with hypertension and diabetes compared to those with a good outcome. Current smoking was higher in patients with a good outcome than among those with a poor outcome.

Table 4 shows differences in clinical characteristics for patients according to BMI. We found that age was inversely associated with BMI and that male sex was more prevalent in the overweight group than in the other BMI groups. Hyperlipidemia and current smoking were less frequent, and cardioembolism was more frequent in the underweight group than in the normal weight, overweight, and obese groups.

Table 5 shows the results of BLR analyses for predictors of a poor outcome at 3 months. Being underweight was a significant predictor for poor outcomes in multivariate analysis. However, obesity had a beneficial effect on the outcome at 3 months after adjusting for multiple covariates. Moreover, addition logistic regression analysis considering age*BMI interaction and entering age and BMI as a continuous variable were not different from the original BLR analysis (Appendix A).

For fp-growth algorithms, we set minimum support bound of 0.04 and a confidence bound of 0.8, according to the average of all possible association rules that could be generated in our dataset. With this parameter setting, 329 rules were generated in total. After a pruning procedure for reducing redundant association rules, five association rules satisfied the support and confidence limits (Table 6). In the top five rules with high confidence & lift values, obesity, age less than 55 years, and mild stroke severity were presented to have a relationship with a good outcome. Figure 2A depicts the relationship of these association rules according to support, confidence, and lift values. Obese was associated with a good outcome, in which no diabetes, mild stroke severity, no hyperlipidemia, male, smoking, and younger age (<55 years) were concurrently observed. Figure 2B shows the coordination plot of significant association rules in our fp-growth algorithm. The rule of being obese, younger, and presenting with no diabetes, no hyperlipidemia, and mild stroke severity was associated with a good outcome with the highest lift value in our rules. Appendix A shows the result of the five most frequent association rules in each BMI subgroup. Underweight patients with severe stroke severity and older age were especially associated with poor outcomes.

## 4. Discussion

In the present study, logistic regression analyses in the established ischemic stroke population revealed that being obese was associated with a good outcome. However, obese patients had a lower age, male gender, and were less likely to present with cardioembolism. In other words, the obesity paradox could be partially explained by the fact that obese patients may be younger than patients with normal BMI or may be more likely to have a stroke that is associated with a non-cardioembolism mechanism.

Welton et al. conducted a prospective investigation of 8528 patients with diabetes and reported that mortality among obese patients was less than among patients with a normal BMI [31]. However, this tendency decreased when the fitness level was high in obese patients, and the authors suggested that the fitness level measured by the metabolic equivalent of the task may be a moderator of the association between obesity and mortality. Meanwhile, age was different for each BMI group in the present study, and notably, the mean age was lowest in the obese group (BMI > 30 kg/m^2^). Bhaskaran et al. analyzed the data of 3.6 million participants from the UK general population for associations between BMI and mortality using prospective survival analysis, and they suggested that mortality was higher in subjects with low (<18.5 kg/m^2^) or high BMI (≥25 kg/m^2^) [32]. The average age was 37 years in their study, and the overweight or obese group was found to be 10 years older than the normal BMI group. Thus, age can be an important factor for analyzing obesity and stroke outcomes. Khan et al. analyzed the relationship between obesity and CVD using individual patient data from ten population-based cohorts [17]. The increase in BMI at cohort enrollment accelerated CVD onset, which eventually increased the longevity of patients with CVD. In addition, high BMI increased CVD mortality compared to non-CVD death. This study followed 19,000 patients for over 50 years overall and was stratified according to age, sex, and BMI status. This approach may have minimized the selection bias, such as lead-time bias or survivorship bias.

After a report regarding the obesity paradox in stroke patients, there has not been a clear explanation for the relationship between positive stroke outcomes and high BMI status [10]. Results of prospective cohort studies within the general population consistently show that high BMI increases stroke risk [6,7,8,9]. However, the contradicting results found within the obesity paradox for stroke have been reported in studies with established stroke cohorts [10,11,12,13,14,15,16,33,34,35,36,37]. Several studies have also reported that the strength of correlation between high BMI and low stroke mortality was attenuated by age [10,13]. An additional consideration is that those with a high BMI group in the established stroke population are consistently reported to be about 3–7 years younger than those with normal BMI in these observational stroke cohorts. According to the Danish Stroke Register report, higher BMI is associated with accelerated ischemic stroke occurrence [38]. Andersen et al. reported that obesity was associated with lower mortality and risk of readmission for recurrent stroke than the normal weight group after applying Cox proportional hazard models [14]. However, concerns regarding the proportional hazard assumption and age discrepancies between each BMI group have not been discussed in detail. Results from the Atherosclerosis Risk in Communities study, which adjusted for time-varying covariates using the G-estimation method, reported that the association between stroke outcome and BMI status could vary due to mishandling time-varying confounders [9]. Hence, age is the most important variable for explaining the relationship between BMI and stroke outcome. In addition, from the results of the logistic regression model, it can be suggested that high BMI is related to a good outcome, but residual confounding factors related to age should be considered when interpreting these associations.

The ARM algorithm was initially designed to determine the specific purchasing patterns of people in the market and to analyze customers’ behavior [18]. Since then, studies using ARM to find valuable linkages among various factors have been reported in the medical field. Szalkai et al. analyzed factors affecting cognitive decline below 15 points in the Mini-Mental Status Examination score within more than 5000 Alzheimer disease databases [39]. Factors such as high aspartate aminotransferase and high serum sodium were identified as significant rules showing a strong association with cognitive decline. In addition, ARM can specify which non-hierarchical phenotype has been achieved by effective clustering of multiple diseases [40,41,42]. However, ARM usually deals with Boolean data, so it is difficult to analyze associations among the numerical data. Further, the antecedents and consequents in an ARM analysis are not a way to depict a potential causal relationship because variables with high proportions are usually applied as antecedents.

Logistic regression analysis represents the individual risk as an odds ratio; hence, it focuses on individual risk [43]. However, ARM evaluates the entire dataset on a microscopic level and can find important patterns in the groups of interest, though the effect sizes are small [44]. To be specific, when logistic regression and ARM analyze the data, these methods can reach a similar result. However, when it comes to the interpretation of the associations between the predictors and response, logistic regression predicts a probability of dependent variable using all the independent variables and significance of a single predictor. ARM methodology finds association rules between the subset of predictors and response with its frequency and significance. As in the Appendix A, for clarification of the result, an interaction term among the age and BMI is considered in the logistic regression to account for an additional effect between predictors on a dependent variable, but it presented no significant relationship with the response. In addition, ARM does not include prior hypotheses for statistical testing and needs to consider interactions among the variables of interest [45]. In other words, ARM is complementary to the conventional logistic regression model. Therefore, if we implement ARM in addition to logistic regression analysis in evaluating the relationship between disease and risk factors, we can make a more explanatory hypothesis by checking the macroscopic and microscopic associations.

The present study used the data of a representative stroke population. However, our study has a few limitations. First, the results did not include numerical laboratory data such as low-density lipoprotein or glucose levels because of the innate disadvantages of the ARM algorithm. Second, a single-center, retrospective observation is prone to have a selection bias, so the results may not be generalizable to the entire stroke population. Third, we used BMI values at the time of admission for evaluating the degree of obesity; however, BMI is well-known to be a time-varying covariate. Therefore, our study does not provide results on how changes in BMI after stroke affect the outcome at 3 months. Finally, we used Asian Pacific World Health Organization criteria for the obesity categorization. Because BMI depends on age, gender, and ethnicity, it is difficult to generalize our findings to all ischemic stroke patients. Therefore, for an international comparison of the impact of BMI on the prognosis of stroke, we should consider which criteria were used for the BMI categorization. To accurately understand the impact of BMI on CVD in AIS, we need later prospective studies considering age stratification and temporal variation of BMI in the research design and analytic phase.

## 5. Conclusions

Our findings using logistic regression analysis suggest that obesity is associated with a good outcome after stroke. However, the results of ARM analysis revealed that being obese was associated with good outcomes by way of younger age at the onset and mild stroke severity. This suggests that the good outcome in AIS patients was not because the patients were obese, but rather because those patients were younger and their stroke severity was less severe than those with a normal BMI. Thus, the ARM algorithm can be used to find novel and valuable linkages among risk factors and outcomes in the medical research field.

## Figures and Tables

**Figure 1 jpm-12-00016-f001:**
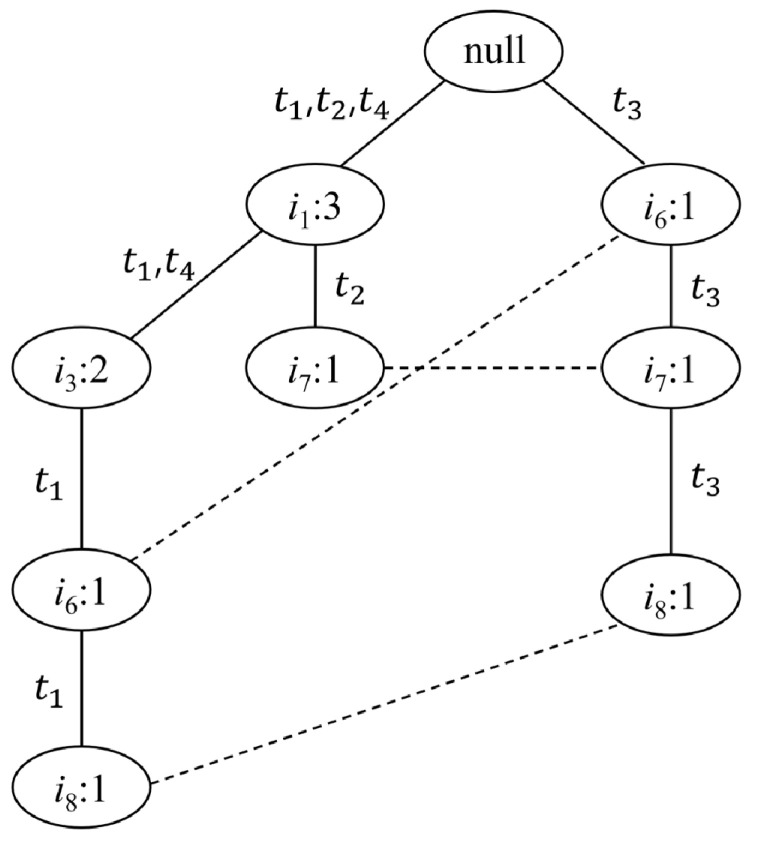
Schematic representation of the frequent pattern growth algorithm with an illustration of fully grown frequent pattern tree using transaction data. *t*, transactions; *i*, item.

**Figure 2 jpm-12-00016-f002:**
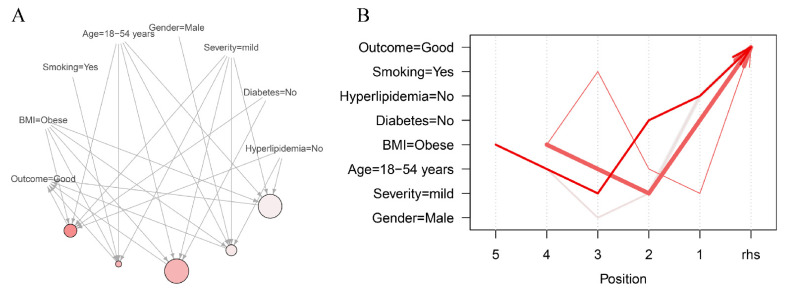
Visualization of significant association rules in patients with acute ischemic stroke, via fp-growth algorithm with minimum support (0.04) and confidence (0.8). (**A**) Network of significant association rules. (**B**) Parallel coordination plot for significant association rules. In the Figure 2A,B, thickness and redness of the lines means support and confidence of the association rules. BMI, body mass index; CE, cardioembolism; rhs, right hand side.

**Table 1 jpm-12-00016-t001:** Definition of formula and explanation of support, confidence, and lift.

	Formula	Definition & Meaning
Support	sup(X→Y)=p(X∩Y)	The value of support means how frequent this rule is appearing in the data.
Confidence	conf(X→Y)=p(Y|X)	The confidence indicates how much the rule is accurate.
Lift	lift(X→Y)=p(Y|X)/p(Y)	The lift measures the dependency between the predictor and the response. The value of lift close to zero indicates independence.

Let X as a subset of predictors and Y as a response. p for probability of an association.

**Table 2 jpm-12-00016-t002:** Example of ordinal and reordered transaction table.

Original Items	Reordered Frequent Items
$t_{1}$$=\{i_{1}$$,\;i_{3}$$,\;i_{5}$$,\;i_{6}$$,\;i_{8}$}	$t_{1}$$=\{i_{1}$$:\;3,\;i_{3}$$:\;2,\;i_{6}$$:\;2,\;i_{8}$: 2}
$t_{2}$$=\{i_{1}$$,\;i_{7}$$,\;i_{9}$}	$t_{1}$$=\{i_{1}$$:\;3,\;i_{7}$: 2}
$t_{3}$$=\{i_{6}$$,\;i_{7}$$,\;i_{8}$}	$t_{1}$$=\{i_{6}$$:\;2,\;i_{7}$$:\;2,\;i_{8}$: 2}
$t_{4}$$=\{i_{1}$$,\;i_{2}$$,\;i_{3}$$,\;i_{4}$}	$t_{1}$$=\{i_{1}$$:\;3,\;i_{3}$: 2}

: denotes the frequency of items in transaction table. t, transactions; i, item.

**Table 3 jpm-12-00016-t003:** Comparison of baseline characteristics of the participants.

	Good Outcome (N = 1671)	Poor Outcome (N = 909)	*p*
Age, years			<0.001
18–54	441 (24.6%)	120 (13.2%)	
55–64	401 (24.0%)	70 (7.8%)	
≥65	859 (51.4%)	719 (79.1%)	
BMI, kg/m^2^			<0.001
Underweight (<18.5)	47 (2.8%)	71 (7.8%)	
Normal weight (18.5–22.9)	540 (32.3%)	359 (39.5%)	
Overweight (23.0–24.9)	463 (27.7%)	220 (24.2%)	
Obese (≥25)	621 (37.2%)	259 (28.5%)	
Stroke Severity, NIHSS			<0.001
Mild (0–5)	1534 (91.8%)	479 (52.7%)	
Moderate (6–14)	100 (6.0%)	226 (24.9%)	
Severe (>14)	37 (2.2%)	204 (22.4%)	
Men	1068 (63.9%)	457 (50.3%)	<0.001
Hypertension	947 (56.7%)	624 (68.6%)	<0.001
Diabetes	446 (26.7%)	319 (35.1%)	<0.001
Current Smoking	578 (34.6%)	184 (20.2%)	<0.001
Cardioembolism	292 (17.5%)	245 (27.0%)	<0.001
Thrombolysis	138 (8.3%)	135 (14.9%)	<0.001
Hyperlipidemia	293 (17.5%)	156 (17.2%)	0.854

BMI, body mass index; NIHSS, National Institute of Health Stroke Scale.

**Table 4 jpm-12-00016-t004:** Differences in clinical characteristics of the patients stratified by body mass index.

	Underweight (N = 118)	Normal Weight (N = 899)	Overweight (N = 683)	Obese (N = 880)	*p*
Age, years					<0.001 ^†^
18–54 years	13 (11.0%)	142 (15.8%)	110 (16.1%)	206 (23.4%)	
55–64 years	12 (10.2%)	162 (18.0%)	169 (24.7%)	188 (21.4%)	
≥65 years	93 (78.8%)	595 (66.2%)	404 (59.2%)	486 (55.2%)	
Stroke Severity, NIHSS					<0.001
Mild (0–5)	70 (59.3%)	658 (73.2%)	550 (80.5%)	735 (83.5%)	
Moderate (6–14)	25 (21.2%)	131 (14.6%)	76 (11.1%)	94 (10.7%)	
Severe (>14)	23 (19.5%)	110 (12.2%)	57 (8.3%)	51 (5.8%)	
Stroke Mechanism					<0.001
CE	39 (33.1%)	220 (24.5%)	130 (19.0%)	148 (16.8%)	
Non-CE	79 (66.9%)	679 (75.5%)	553 (81.0%)	732 (83.2%)	
Outcome at 3 months					<0.001
Good (mRS score 0–2)	47 (39.8%)	540 (60.1%)	463 (67.8%)	621 (70.6%)	
Poor (mRS score 3–6)	71 (60.2%)	359 (39.9%)	220 (32.2%)	259 (29.4%)	
Men	49 (41.5%)	495 (55.1%)	432 (63.3%)	549 (62.4%)	<0.001
Hypertension	67 (56.8%)	492 (54.7%)	200 (29.3%)	281 (31.9%)	<0.001
Hyperlipidemia	9 (7.6%)	146 (16.2%)	113 (16.5%)	181 (20.6%)	0.002
Current Smoking	19 (16.1%)	268 (29.8%)	201 (29.4%)	274 (31.1%)	0.010
Diabetes	37 (31.4%)	247 (27.5%)	200 (29.3%)	281 (31.9%)	0.217
Thrombolysis	12 (10.2%)	101 (11.2%)	70 (10.2%)	90 (10.2%)	0.891

^†^*p* for analysis of variances. Statistical significance of cate variables is represented with the *p*-values of the χ^2^ test. NIHSS, National Institute of Health Stroke Scale; CE, cardioembolic; mRS, modified Rankin Scale score.

**Table 5 jpm-12-00016-t005:** Results of binary logistic regression analysis for predictors of poor functional outcome at 3 months in patients with acute ischemic stroke.

	Univariate OR (95% CI)	*p*	Multivariate OR (95% CI)	*p*
Age, years				
18–54	0.29 (0.24–0.36)	0.001	0.63 (0.44–0.91)	0.013
55–64	1.00 (reference)	-	1.00 (reference)	-
≥65	2.87 (2.29–3.59)	<0.001	2.26 (1.74–2.93)	<0.001
BMI, kg/m^2^				
Underweight (<18.5)	2.27 (1.54–3.36)	<0.001	1.69 (1.07–2.66)	0.024
Normal weight (18.5–22.9)	1.00 (reference)	-	1.00 (reference)	-
Overweight (23.0–24.9)	0.71 (0.58–0.88)	0.001	0.82 (0.64–1.05)	0.119
Obese (≥25)	0.63 (0.52–0.76)	<0.001	0.78 (0.62–0.99)	0.041
Stroke Severity, NIHSS				
Mild (0–5)	0.14 (0.11–0.18)	<0.001	0.11 (0.08–0.15)	<0.001
Moderate (6–14)	1.00 (reference)	-	1.00 (reference)	-
Severe (>14)	2.44 (1.60–3.72)	<0.001	2.37 (1.51–3.70)	<0.001
Diabetes	1.49 (1.25–1.77)	<0.001	1.61 (1.30–1.99)	<0.001
Current Smoking	0.48 (0.40–0.58)	<0.001	0.67 (0.52–0.86)	0.001
Thrombolysis	1.94 (1.51–2.49)	<0.001	0.64 (0.45–0.91)	0.012
Cardioembolism	1.74 (1.44–2.11)	<0.001	0.75 (0.58–0.96)	0.024
Hypertension	1.67 (1.41–1.98)	<0.001	1.14 (0.92–1.41)	0.228
Hyperlipidemia	0.97 (0.79–1.21)	0.811	0.90 (0.70–1.17)	0.444
Men	0.57 (0.48–0.67)	<0.001	0.92 (0.75–1.14)	0.854

OR, odds ratio; CI, confidence interval; BMI, body mass index; NIHSS, National Institute of Health Stroke Scale.

**Table 6 jpm-12-00016-t006:** Results of the frequent pattern growth algorithm of independent and dependent variables in patients with acute ischemic stroke.

	LHS	RHS	Support	Confidence	Lift	Count
1	{Hyperlipidemia = No, Diabetes = No, Severity = mild, Age = 18–54 years, BMI = Obese}	Good	0.0488	0.9767	1.5080	126
2	{Severity = mild, Age = 18–54 years, Smoking = Yes, BMI = Obese}	Good	0.0434	0.9739	1.5037	112
3	{Diabetes = No, Severity = mild, Age = 18–54 years, BMI = Obese}	Good	0.0577	0.9738	1.5036	149
4	{Hyperlipidemia = No, Severity = mild, Gender = Male, Age = 18–54 years, BMI = Obese}	Good	0.0472	0.9682	1.4949	122
5	{Hyperlipidemia = No, Severity = mild, Age = 18–54 years, BMI = Obese}	Good	0.0573	0.9673	1.4935	148

LHS, left hand side; RHS, right hand side; BMI, body mass index; CE, cardioembolism.

## Data Availability

The data presented in this study are available on request from the corresponding author. The data are not publicly available due to the policy of our IRB.

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
