# Peer review of "Another Look at Obesity Paradox in Acute Ischemic Stroke: Association Rule Mining"

_jpm, 2021, doi:10.3390/jpm12010016_

Round 1
Reviewer 1 Report
COMMENTS TO THE MANUSCRIPT ‘ANOTHER LOOK AT THE OBESITY PARADOX IN ACUTE ISCHEMIC STROKE: ASSOCIATION RULE MINING’.
The subject the authors have studied represents an interesting topic. The paper is well structured and presented, the English is good. The paper has scientific interest.
ABSTRACT
Line 26-27: ‘We aimed to evaluate and compare the obesity paradox using binary logistic regression…’
It would be better to state that they compare the two methods in assessing the obesity paradox. ‘Compare the obesity paradox’ - with what?
INTRODUCTION
I recommend the authors to reformulate their research goal also here. They state ‘we aimed to assess and compare the relationship between obesity and ischemic stroke outcome’ using logistic regression and the ARM method. The actual outcome of ischemic stroke is a real clinical result that is not depending on the method of measurement. What they have studied, is whether two different statistical methods give us differing assessment of the clinical result. They compared two methods.
Given this basis for my review, I expect to learn whether ARM adds useful information or even may be better than logistic regressions. This assumes the logistic regression is applied optimally.
MATERIALS AND METHODS
2.1. Participants.
The study population seems to be a subgroup recruited from a larger cohort (stroke registry). It would be useful to specify the selection criteria sharper. They state that patients were admitted to the hospital within seven days after onset of focal neurologic deficits. Is this related to inclusion in the registry they have recruited patients from or was it a criterium for recruiting from the registry to this current study. In that case, did they use a combination of ‘admittance less than seven days after onset’ AND ‘relevant acute lesions on diffusion MRI’? The reason I ask this is that there is a low contingent of thrombolysis and embolectomy, procedures that usually are related to time of admission after onset of symptoms. Furthermore, as they have recruited during an eight-year period, it should also be stated whether such procedures or other parts of the treatment were modified/changed/increased during the recruitment period as such interventions have significant influence on outcomes on and are given to suitable patients, a possible ‘selection bias’.
2.1.1. Measurement of body mass index.
The authors have applied Asian Pacific World Health Organization criteria which is of course appropriate for the study group considered, but such criteria show some variation across people from different populations, geographical locations and descent. It is also well known that BMI is depending on age and gender, and that Asian people have a higher range of percent body fat for the same level of BMI than Caucasian people. Accordingly, the optimal cut-off for obesity in Asian populations could be different to that of Western subjects. This is not discussed in the paper.
2.2/3. Statistical analysis and Association rule mining.
The ARM method is sufficiently described for an analyst experienced with the method. But I would recommend an initial presentation of what ARM is if the authors want to get “less ARM-skilled” readers on board. I suppose (but may be wrong) that terms as Support, Confidence and Lift in this setting is not known to a large public.
For example, from line 132 insert a simple table or panel illustrating:
ARM is based on …..
Support: Explain formula/definition/meaning
Confidence: Explain formula/definition/meaning
Lift: Explain formula/definition/meaning
and so on.
Logistic regression.
I have a major concern related to the LR. The authors state in their conclusion (line 340 +):
‘Our findings using logistic regression analysis suggest that obesity is associated with a good outcome after stroke.’
This is correct based on the LR they performed if they add ‘using univariate regressions’ (I observe the results in Table 3 as univariate analyses).
But I am quite worried related to their univariate analyses. If we consider their two groups of ‘Good’ and ‘Poor’ and the outcome across the weight classes, we see that the percent ‘Good outcome’ in each BMI class is 3%, 32%, 28 % and 37 % respectively. Then, 37 % in obesity class versus 32 % in normal is not clinically impressing. Does it have any clinical significance? Similarly, considering the outcomes within each BMI class of normal, overweight and obesity, ‘percent good outcome’ is 67,8%, 70,6 % and 60.1 % respectively. I believe that such numbers strongly advertise that using only univariate analyses is at risk.
Fortunately, LR is a method to assess a dichotomous outcome to analyze binary responses to continuous variables. Not having the database available I may be misled, but I miss an LR analysis using the variables such as BP, BMI and age as continuous variables, thus eliminating the obvious pitfalls that may be induced by the categorizations at given cut points. And possibly including “Thrombolysis” and ‘Embolism’ as binary variables to control for these, possibly dummy variables for year to control for time. I understand that categorizations must be done for the ARM. But LR should also be performed using continuous variables including covariates across the differing variables. What would be the effect of including age* BMI? This might possibly have disclosed that LR could detect similar linkages as ARM and ‘kill’ the paradox with LR as well? From my own experience on LR, I actually believe that the authors have missed rather interesting aspects of their data and suggest looking into the mentioned perspectives.
DISCUSSION
Lines 315 and following.
I refer to my previous section. Although LR represents individual risks and ARM may represent a macroscopic perspective, I am not sure this is of high interest. I find the statement ‘ARM algorithm can be used to find novel and valuable linkages among risk factors and outcomes in the medical research field’ only partly true. The 'novelty' statement depends on the comparison, which should be based on more thorough LR application.
The ‘proof of concept’ of the ARM may be interesting, but whether it adds any value to LR in the current setting is not convincingly documented when compared to the chosen LR. I find ARM an interesting method, in part mimicking some AI and deep learning principles, but before ‘we implement ARM in addition to LR’ (line 322), LR should be more thoroughly analyzed.
In the section line 326 +, the authors discuss some limitations of their study. This could also include the limitations related to their application of the Asian Pacific World Health Organization criteria, as this may possibly influence the applicability for the results on other populations/groups.
Author Response
Reviewer #1.
1) The subject the authors have studied represents an interesting topic. The paper is well structured and presented, the English is good. The paper has scientific interest.
ABSTRACT
Line 26-27: ‘We aimed to evaluate and compare the obesity paradox using binary logistic regression…’
It would be better to state that they compare the two methods in assessing the obesity paradox. ‘Compare the obesity paradox’ - with what?
Response 1) Thank you for the comment. We acknowledge that the statement in the abstract section was not enough to provide a clear understanding, so we amended this point in the manuscript as follows.
“We aimed to verify the existing obesity paradox through binary logistic regression (BLR) and clarify the paradox via association rule mining (ARM)”
2) INTRODUCTION
I recommend the authors to reformulate their research goal also here. They state ‘we aimed to assess and compare the relationship between obesity and ischemic stroke outcome’ using logistic regression and the ARM method. The actual outcome of ischemic stroke is a real clinical result that is not depending on the method of measurement. What they have studied, is whether two different statistical methods give us differing assessment of the clinical result. They compared two methods.
Given this basis for my review, I expect to learn whether ARM adds useful information or even may be better than logistic regressions. This assumes the logistic regression is applied optimally.
Response 2) Similar to the above amendments in the abstract section, we amended this point in the manuscript as follows.
"Therefore, we aimed to verify the obesity paradox with logistic regression and provide a new perspective through ARM method on the relationship between obesity and ischemic stroke outcome”
3) MATERIALS AND METHODS
2.1. Participants.
The study population seems to be a subgroup recruited from a larger cohort (stroke registry). It would be useful to specify the selection criteria sharper. They state that patients were admitted to the hospital within seven days after onset of focal neurologic deficits. Is this related to inclusion in the registry they have recruited patients from or was it a criterium for recruiting from the registry to this current study. In that case, did they use a combination of ‘admittance less than seven days after onset’ AND ‘relevant acute lesions on diffusion MRI’? The reason I ask this is that there is a low contingent of thrombolysis and embolectomy, procedures that usually are related to time of admission after onset of symptoms. Furthermore, as they have recruited during an eight-year period, it should also be stated whether such procedures or other parts of the treatment were modified/changed/increased during the recruitment period as such interventions have significant influence on outcomes on and are given to suitable patients, a possible ‘selection bias’.
Response 3) Thank you for the comment. We added a supplemental figure for the inclusion and exclusion criteria for this study. Our stroke registry is a part of the nationally-representative stroke registry in our country. We mainly collect the data for acute ischemic stroke. However, there is a few data for hemorrhagic stroke or those with late-onset (>7 days of symptom onset). During the data capture period, procedures affecting the prognosis of patients, such as extending the time window of intravenous thrombolysis and endovascular treatment, were reflected in all included patients.
4) 2.1.1. Measurement of body mass index.
The authors have applied Asian Pacific World Health Organization criteria which is of course appropriate for the study group considered, but such criteria show some variation across people from different populations, geographical locations and descent. It is also well known that BMI is depending on age and gender, and that Asian people have a higher range of percent body fat for the same level of BMI than Caucasian people. Accordingly, the optimal cut-off for obesity in Asian populations could be different to that of Western subjects. This is not discussed in the paper.
Response 4) Thank you for the comment. Because the subject of our study is a comparison of logistic regression and ARM for the impact of obesity according to the BMI class rather than the different criteria of obesity. Therefore, we added the following to the limitation section instead of the detailed comparison for the definition of obesity criteria.
Finally, we used Asian Pacific World Health Organization criteria for the obesity categorization. Because BMI depends on age, gender, and ethnicity, it is difficult to generalize our findings to all ischemic stroke patients. Therefore, for international comparison of the impact of BMI on the prognosis of stroke, we should consider which criteria were used for the BMI categorization.
5) 2.2/3. Statistical analysis and Association rule mining.
The ARM method is sufficiently described for an analyst experienced with the method. But I would recommend an initial presentation of what ARM is if the authors want to get “less ARM-skilled” readers on board. I suppose (but may be wrong) that terms as Support, Confidence and Lift in this setting is not known to a large public.
For example, from line 132 insert a simple table or panel illustrating:
ARM is based on …..
Support: Explain formula/definition/meaning
Confidence: Explain formula/definition/meaning
Lift: Explain formula/definition/meaning
and so on.
Response 5) Thank you for the comment. As in the review, we added a formula/definition/meaning with Table 1 as follows.
Table 1. Definition of formula and explanation of support, confidence, and lift.
|
|
Formula |
Definition & Meaning |
|
Support |
The value of support means how frequent this rule is appearing in the data |
|
|
Confidence |
The confidence indicates how much the rule is accurate. |
|
|
Lift |
The lift measures the dependency between the predictor and the response. The value of lift close to zero indicates the independence. |
Let X as a subset of predictors and Y as a response.
6) I have a major concern related to the LR. The authors state in their conclusion (line 340 +):
‘Our findings using logistic regression analysis suggest that obesity is associated with a good outcome after stroke.’
This is correct based on the LR they performed if they add ‘using univariate regressions’ (I observe the results in Table 3 as univariate analyses).
But I am quite worried related to their univariate analyses. If we consider their two groups of ‘Good’ and ‘Poor’ and the outcome across the weight classes, we see that the percent ‘Good outcome’ in each BMI class is 3%, 32%, 28 % and 37 % respectively. Then, 37 % in obesity class versus 32 % in normal is not clinically impressing. Does it have any clinical significance? Similarly, considering the outcomes within each BMI class of normal, overweight and obesity, ‘percent good outcome’ is 67,8%, 70,6 % and 60.1 % respectively. I believe that such numbers strongly advertise that using only univariate analyses is at risk.
Response 6) Thank you for the detailed advice. First, the model originally proposed in our document was a multivariate logistic regression model which is already accounting for the correlation between predictors. With the nice tip in the review, we found it will be a more clear response if we expand our Table 4 including the result of univariate logistic regression with our multivariate model. Therefore we amend it accordingly in the methods as follows with Table 4.
Table 4. Results of binary logistic regression analysis for predictors of poor functional outcome at 3 months in patients with acute ischemic stroke.
|
|
Univariate OR (95% CI) |
P |
Multivariate OR (95% CI) |
P |
|
Age, years |
|
|
|
|
|
18-54 |
0.29 (0.24-0.36) |
0.001 |
0.63 (0.44-0.91) |
0.013 |
|
55-64 |
1.00 (reference) |
- |
1.00 (reference) |
- |
|
65 |
2.87 (2.29-3.59) |
<0.001 |
2.26 (1.74-2.93) |
<0.001 |
|
BMI, kg/m2 |
|
|
|
|
|
Underweight (<18.5) |
2.27 (1.54-3.36) |
<0.001 |
1.69 (1.07-2.66) |
0.024 |
|
Normal weight (18.5-22.9) |
1.00 (reference) |
- |
1.00 (reference) |
- |
|
Overweight (23.0-24.9) |
0.71 (0.58-0.88) |
0.001 |
0.82 (0.64-1.05) |
0.119 |
|
Obese (25) |
0.63 (0.52-0.76) |
<0.001 |
0.78 (0.62-0.99) |
0.041 |
|
Stroke Severity, NIHSS |
|
|
|
|
|
Mild (0-5) |
0.14 (0.11-0.18) |
<0.001 |
0.11 (0.08-0.15) |
<0.001 |
|
Moderate (6-14) |
1.00 (reference) |
- |
1.00 (reference) |
- |
|
Severe (>14) |
2.44 (1.60-3.72) |
<0.001 |
2.37 (1.51-3.70) |
<0.001 |
|
Diabetes |
1.49 (1.25-1.77) |
<0.001 |
1.61 (1.30-1.99) |
<0.001 |
|
Current Smoking |
0.48 (0.40-0.58) |
<0.001 |
0.67 (0.52-0.86) |
0.001 |
|
Thrombolysis |
1.94 (1.51-2.49) |
<0.001 |
0.64 (0.45-0.91) |
0.012 |
|
Cardioembolism |
1.74 (1.44-2.11) |
<0.001 |
0.75 (0.58-0.96) |
0.024 |
|
Hypertension |
1.67 (1.41-1.98) |
<0.001 |
1.14 (0.92-1.41) |
0.228 |
|
Hyperlipidemia |
0.97 (0.79-1.21) |
0.811 |
0.90 (0.70-1.17) |
0.444 |
|
Men |
0.57 (0.48-0.67) |
<0.001 |
0.92 (0.75-1.14) |
0.854 |
OR, odds ratio; CI, confidence interval; BMI, body mass index; NIHSS, National Institute of Health Stroke Scale.
7) Fortunately, LR is a method to assess a dichotomous outcome to analyze binary responses to continuous variables. Not having the database available I may be misled, but I miss an LR analysis using the variables such as BP, BMI and age as continuous variables, thus eliminating the obvious pitfalls that may be induced by the categorizations at given cut points. And possibly including “Thrombolysis” and ‘Embolism’ as binary variables to control for these, possibly dummy variables for year to control for time. I understand that categorizations must be done for the ARM. But LR should also be performed using continuous variables including covariates across the differing variables.
Response 7) Thank you for the comment, that stratified variables can affect the variance of predictors in the regression which can lead to the significant change of coefficients in the regression analysis. As in your advice, we added a new supplementary table (Supplementary Table S2), which is the results of univariate and multivariate binary logistic regressions using continuous age and BMI, showing there are no significant changes even after stratifying the aforementioned variables.
Supplementary Table S2. Results of binary logistic regression analysis for predictors of poor functional outcome at 3 months in patients with acute ischemic stroke, considering the variable age and BMI as continuous values.
|
|
Univariate OR (95% CI) |
P |
Multivariate OR (95% CI) |
P |
|
Age, years |
1.06 (1.05-1.07) |
<0.001 |
1.96 (1.74-2.21) |
<0.001 |
|
BMI, kg/m2 |
0.92 (0.90-0.94) |
<0.001 |
0.92 (0.83-1.01) |
0.090 |
|
Stroke Severity, NIHSS |
|
|
|
|
|
Mild (0-5) |
0.14 (0.11-0.18) |
<0.001 |
0.11 (0.08-0.15) |
<0.001 |
|
Moderate (6-14) |
1.00 (reference) |
- |
1.00 (reference) |
- |
|
Severe (>14) |
2.44 (1.60-3.72) |
<0.001 |
2.45 (1.56-3.86) |
<0.001 |
|
Diabetes |
1.49 (1.25-1.77) |
<0.001 |
1.69 (1.37-2.10) |
<0.001 |
|
Current Smoking |
0.48 (0.40-0.58) |
<0.001 |
0.71 (0.55-0.91) |
0.006 |
|
Thrombolysis |
1.94 (1.51-2.49) |
<0.001 |
0.69 (0.49-0.98) |
0.039 |
|
Cardioembolism |
1.74 (1.44-2.11) |
<0.001 |
0.71 (0.55-0.92) |
0.009 |
|
Hypertension |
1.67 (1.41-1.98) |
<0.001 |
1.05 (0.85-1.31) |
0.650 |
|
Hyperlipidemia |
0.97 (0.79-1.21) |
0.811 |
0.91 (0.70-1.18) |
0.466 |
|
Men |
0.57 (0.48-0.67) |
<0.001 |
0.98 (0.79-1.22) |
0.870 |
OR, odds ratio; CI, confidence interval; BMI, body mass index; NIHSS, National Institute of Health Stroke Scale.
8) What would be the effect of including age* BMI? This might possibly have disclosed that LR could detect similar linkages as ARM and ‘kill’ the paradox with LR as well? From my own experience on LR, I actually believe that the authors have missed rather interesting aspects of their data and suggest looking into the mentioned perspectives.
Response 8) Thank you for the new perspective on our regression model, I believe this will help us to prove that we analyzed our data in various perspectives via regression. The use of interaction, in our regression model, is to account for the additional effect induced by the relationship between BMI and age on the outcome. By assuming there is an interaction effect from the BMI and age on the outcome, we constructed a new binary logistic model as in the following Supplementary Table S3.
Supplementary Table S3. Results of binary logistic regression analysis for predictors, considering the interaction between the stratified age and BMI, of poor functional outcome at 3 months in patients with acute ischemic stroke.
|
|
Multivariate OR (95% CI) |
P |
|
|
Age, years |
|
|
|
|
18-54 |
0.77 (0.41-1.44) |
0.4062 |
|
|
55-64 |
1.00 (reference) |
- |
|
|
65 |
2.78 (1.76-4.39) |
<0.001 |
|
|
BMI, kg/m2 |
|
|
|
|
Underweight (<18.5) |
1.14 (0.27-4.69) |
0.8602 |
|
|
Normal weight (18.5-22.9) |
1.00 (reference) |
- |
|
|
Overweight (23.0-24.9) |
0.93 (0.52-1.68) |
0.8212 |
|
|
Obese (25) |
1.18 (0.68-2.05) |
0.5614 |
|
|
Stroke Severity, NIHSS |
|
|
|
|
Mild (0-5) |
0.11 (0.08-0.15) |
<0.001 |
|
|
Moderate (6-14) |
1.00 (reference) |
- |
|
|
Severe (>14) |
2.33 (1.49-3.64) |
<0.001 |
|
|
Diabetes |
1.62 (1.31-2.01) |
<0.001 |
|
|
Current Smoking |
0.68 (0.53-0.87) |
0.0021 |
|
|
Thrombolysis |
0.65 (0.46-0.91) |
0.0138 |
|
|
Cardioembolism |
0.75 (0.58-0.96) |
0.0244 |
|
|
Hypertension |
1.14 (0.92-1.41) |
0.2349 |
|
|
Hyperlipidemia |
0.90 (0.70-1.17) |
0.4248 |
|
|
Men |
0.93 (0.75-1.15) |
0.4793 |
|
|
BMI, kg/m2 |
Age, years |
Interaction OR |
Interaction P |
|
Underweight (<18.5) |
18-54 |
0.99 (0.13-7.45) |
0.9926 |
|
Underweight (<18.5) |
65 |
1.63 (0.36-7.36) |
0.5268 |
|
Overweight (23.0-24.9) |
18-54 |
1.23 (0.49-3.08) |
0.6558 |
|
Overweight (23.0-24.9) |
65 |
0.81 (0.42-1.56) |
0.5325 |
|
Obese (25) |
18-54 |
0.51 (0.21-1.24) |
0.1371 |
|
Obese (25) |
65 |
0.63 (0.53-0.87) |
0.1402 |
OR, odds ratio; CI, confidence interval; BMI, body mass index; NIHSS, National Institute of Health Stroke Scale.
As in the result of Supplementary Table S3, it was not able to see any additional effect of interaction on the response.
9) DISCUSSION
Lines 315 and following.
I refer to my previous section. Although LR represents individual risks and ARM may represent a macroscopic perspective, I am not sure this is of high interest. I find the statement ‘ARM algorithm can be used to find novel and valuable linkages among risk factors and outcomes in the medical research field’ only partly true. The 'novelty' statement depends on the comparison, which should be based on more thorough LR application.
The ‘proof of concept’ of the ARM may be interesting, but whether it adds any value to LR in the current setting is not convincingly documented when compared to the chosen LR. I find ARM an interesting method, in part mimicking some AI and deep learning principles, but before ‘we implement ARM in addition to LR’ (line 322), LR should be more thoroughly analyzed.
Response 9) Thank you for the comment. As in the reviewer’s review, we performed additional analyses and illustrated the results through various regression models. From regressions and ARM, we suggested a firm statement on our ARM analysis that association analysis provides more detailed information of linkages between variables compared with logistic regression model and it needs no assumption on the links. For describing the advantages of the ARM, we amended our written work as below.
“However, ARM evaluates the entire dataset on a microscopic level and can find important patterns in the groups of interest, though the effect sizes are small [43]. To be specific, when logistic regression and ARM analyze the data, these methods can reach a similar result. However, when it comes to the interpretation of the associations between the predictors and response, logistic regression predicts a probability of dependent variable using all the independent variables and significance of a single predictor. However, ARM methodology finds association rules between the subset of predictors and response with its’ frequency and significance. As in the Supplementary Table S3, for clarification of the result, interaction term among the age and BMI is considered in the logistic regression to account for an additional effect between predictors on a dependent variable, but it presented no significant relationship with the response”
10) In the section line 326 +, the authors discuss some limitations of their study. This could also include the limitations related to their application of the Asian Pacific World Health Organization criteria, as this may possibly influence the applicability for the results on other populations/groups.
Response 10) the same response for 8)
Thank you for the comment. Because the subject of our study is a comparison of logistic regression and ARM for the impact of obesity according to the BMI class rather than the different criteria of obesity. Therefore, we added the following to the limitation section instead of the detailed comparison for the definition of obesity criteria.
Finally, we used Asian Pacific World Health Organization criteria for the obesity categorization. Because BMI depends on age, gender, and ethnicity, it is difficult to generalize our findings to all ischemic stroke patients. Therefore, for international comparison of the impact of BMI on the prognosis of stroke, we should consider which criteria were used for the BMI categorization.
Reviewer 2 Report
The study by Pum-Jun Kim et al. is undoubtedly interesting. It was performed based in over 2500 patients with AIS. The authors observed that being obese was associated with the good outcomes by way of young age at onset, mild stroke severity. The study can be improved:
- Please add the information on the place where patients were recruited/hospitalized.
- Instead of blood sugar use blood glucose.
- Please add citation on the definitions of diabetes and hyperlipidemia.
Author Response
The study by Pum-Jun Kim et al. is undoubtedly interesting. It was performed based in over 2500 patients with AIS. The authors observed that being obese was associated with the good outcomes by way of young age at onset, mild stroke severity. The study can be improved:
1) Please add the information on the place where patients were recruited/hospitalized.
Response 1) Thank you for the comment. We added the supplemental figure for the inclusion and exclusion criteria. We included the patients who were admitted (hospitalized) within 7 days of symptom onset.
2) Instead of blood sugar use blood glucose.
Response 2) Thank you for the comment. We amended this properly.
3) Please add citation on the definitions of diabetes and hyperlipidemia.
Response 3) Thank you for the comment. We added the citation for the definition of diabetes and hyperlipidemia, which are represented in our multicenter registry database.